# MENTAL FATIGUE MONITORING USING BRAIN DYNAMICS PREFERENCES

## ABSTRACT

Driver's cognitive state of mental fatigue significantly affects driving performance and more importantly public safety. Previous studies leverage the response time (RT) as the metric for mental fatigue and aim at estimating the exact value of RT using electroencephalogram (EEG) signals within a regression model. However, due to the easily corrupted EEG signals and also non-smooth RTs during data collection, regular regression methods generally suffer from poor generalization performance. Considering that human response time is the reflection of brain dynamics preference rather than a single value, a novel model called Brain Dynamics ranking (BDrank) has been proposed. BDrank could learn from brain dynamics preferences using EEG data robustly and preserve the ordering corresponding to RTs. BDrank model is based on the regularized alternative ordinal classification comparing to regular regression based practices. Furthermore, a transition matrix is introduced to characterize the reliability of each channel used in EEG data, which helps in learning brain dynamics preferences only from informative EEG channels. In order to handle large-scale EEG signals and obtain higher generalization, an online-generalized Expectation Maximum (OnlineGEM) algorithm also has been proposed to update BDrank in an online fashion. Comprehensive empirical analysis on EEG signals from 44 participants shows that BDrank together with OnlineGEM achieves substantial improvements in reliability while simultaneously detecting possible less informative and noisy EEG channels.

## 1 INTRODUCTION

As reported by sleep health report (Adams et al., 2017), mental fatigue is a major cause in $33\% - 45\%$ of all road accidents. In general, mental fatigue (Boksem & Tops, 2008) refers to the inability to maintain optimal cognitive performance in continuous task of the high demand of cognitive activity. Such inability in the context of driver could lead to accidents with severe consequences. Individuals may find themselves in a mental fatigue state because of lack of sleep, continuous driving for long-time, midnight driving, monotonous driving, and driving during the influence of sleeping drugs or sleep disorders (Ji et al., 2004; Ting et al., 2008).

In response to these critical issues, several methods (Jap et al., 2009; Wascher et al., 2014; Cook et al., 2007; Lal et al., 2003; de Naurois et al., 2017) have been proposed to estimate and predict the mental fatigue based on EEG and RT (Fig. 1a). Some of these methods, however, performed considerably well for some participants but failed for others due to lack of robustness. There are several challenges behind such instability and one of such problem is how to use RT effectively. RT is the most resourceful piece of information to predict mental fatigue. However, it is easily affected by the instrumental error, mind wandering or any other task non-related factor. A previous study (Wei et al., 2015) tried to overcome this problem by providing different techniques to adjust RT, they also tried removing outliers nevertheless, failed to make it work for all participants. The regression assumption of this method between EEG signals and RT is not correct. Human's RT is usually the result of preference (Izuma & Adolphs, 2013) in brain dynamics during the task, than just a single value. These preferences of human can be affected by different cognition (Möckel et al., 2015) like mind wandering (Lin et al., 2016), and/or lower level of attention (Chuang et al., 2018). Therefore, the relationship between EEG and RT including the extreme/abnormal RT should be taken care in the way that reflect human brain dynamics preferences by the developed technique itself.

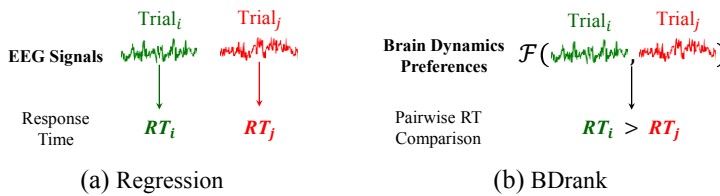

Figure 1: (a) Regression model with EEG signals. (b) BDrank model with brain dynamics preferences.

Another important problem is related to different brain regions, which are normally responsible for different functionalities. There was an attempt to choose different brain regions (Wascher et al., 2014) for a method during mental fatigue prediction but these regions of the brain are not necessary same for all participants (Gramann et al., 2006). For example, existing work (Wascher et al., 2014) used frontal theta to represent a different level of mental fatigue for all participants. In such case learning model's reliability would inevitably degrade because of possibly noisy channels chosen, on different brain regions, by the method. Some previous work (de Naurois et al., 2017), attempted to solve this issue using artificial neural network models but still failed to provide convincing results. Again these mentioned work reflect developed methods should be based on brain dynamics preferences rather than fix model or regions of brain.

To overcome the above-mentioned problems, a new approach has been proposed. We called it BDrank such that brain dynamics ranking. This approach not only learns from brain dynamics preferences for mental fatigue but also other cognitive states (Lal et al., 2003), while effectively preserving the exact ordering of RT (Fig. 1b). This approach surprisingly improved over defects of previous models and their performance due to noisy and extreme RT. Furthermore, BDrank model also proposes to use transition matrix to evaluate the high confidence BDrank (HC-BDrank) sources among different EEG channels, which contributes highly toward task performance. In order to handle large-scale EEG signals and obtain higher generalization, an online-generalized Expectation Maximum (OnlineGEM) (Roche, 2011) algorithm also has been proposed. Comprehensive empirical experiments on EEG signals from 44 participants show that BDrank model together with OnlineGEM algorithm delivers substantial improvements and robust performance while simultaneously providing the information about noisy or less informative EEG channels.

## 2 MATERIALS AND METHODS:

**a) Experiment Paradigm:** This paper utilized the 33-channels EEG data recorded in the previous study (Huang et al., 2015) from 44 adult participants while performing long sustained attention task. The experiment has been conducted using a virtual-reality (VR) dynamic driving simulator (Fig. 2D-E). The task involves driving on the four-lane highway while lane-departure events were randomly induced deviation toward the side of the road from the original position. Each participant was instructed to quickly respond to steer back to original position. A complete trial

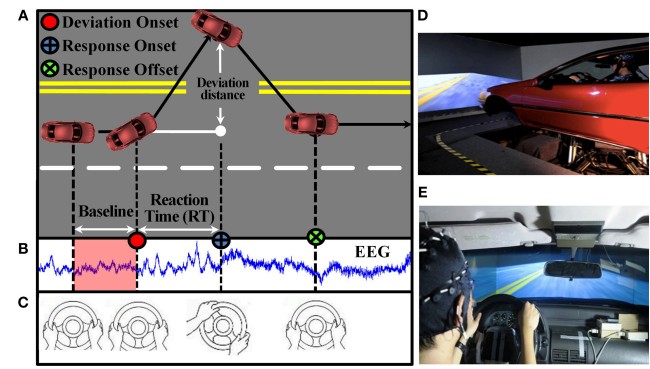

Figure 2: Sustained-attention driving task

in this study (Fig. 2A), includes 10s baseline, deviation onset, response onset, and response offset (Fig. 2B-C). The next trial occurs within an interval of 5-10s after finishing the current trial. Each participant completed $T$ trials within 1.5h. For each trial $i$, the EEG signals $\{x_{n,i}\}_{n=1}^N$ from $N$ different channels were recorded simultaneously and the corresponding response time $RT_i$ was also collected afterwards. If a participant fell asleep during the experiment, there was no feedback to wake him up.

The response time is an intuitive indicator used to assess human mental fatigue. Therefore, the common practice for mental fatigue monitoring is to find a robust mapping for humans' response time (RT) to an emergent situation using the EEG signals recorded beforehand. The natural way to forecast the response time with the EEG signals is to formulate it as a regression task (Fig. 1a), namely finding a nonlinear mapping (e.g. neural network, SVR) from the EEG signals $x$ to the corresponding RT. However, due to the easily corrupted properties of the EEG signals and the existence of extreme values in response times during data collection, focusing on predicting the exact value of a noisy and non-smooth measurement (the response time) is easier to create a near-perfectly fitted model with poor generalization performance (See Table 1 and Fig. 3 for more details). This creates a dilemma: it requires a powerful learning model to predict response time with the complex EEG signals (indeed, it is exactly our target) but it is not so significant to excessively approximate the exact value of response time, especially the extreme values.

Here comes the problem, how to find an efficient way to learn from the noisy response time while the exact value is not so significant? Actually, the RTs are defined in the totally ordered space $R$. The totally ordered space owns its structure meanings, which are preserved by the pairwise comparisons between the RTs. The pairwise comparisons indeed preserve the whole relative structure information between the response times while ignoring their absolute numerical information. Therefore, predicting the orderings of the pairwise comparisons can be accepted as a regularized alternative of previous regression model. Further, it makes it more flexible to consider more powerful learning unit in the proposed model.

Therefore, there is no longer requirement to estimate the RT with a regression model. Instead, the proposed approach will transform it into an ordinal classification problem, and focus on correctly preserving the whole orderings between the pairwise RT comparisons (Fig. 1b). First, the preference propositions[1] could be constructed as follows,

$$\mathscr{D}_1 = \{\rho_m\}_{m=1}^{M_1} = \{RT_{m,1} > RT_{m,2}\}_{m=1}^{M_1}, \qquad \mathscr{D}_2 = \{\rho_{m'}\}_{m'=1}^{M_2} = \{RT_{m',1} \approx RT_{m',2}\}_{m'=1}^{M_2}. \qquad (1)$$

where $M_1$ and $M_2$ denote the number of type-1 and type-2 preference propositions. $\mathscr{D}_1$ denotes the type-1 preference propositions that the orderings between the RTs are significant; $\mathscr{D}_2$ denotes the type-2 preference propositions that the RTs in each comparison are comparable. Then, the brain dynamics preference was constructed for each proposition with the corresponding pairwise EEG signals recorded from each channel. For brevity of notations, new index (the notation in Eq.1) is adopted in the following, instead of the original index used in sustained-attention driving task (Fig. 1). Namely, the $m$-th proposition $\rho_m \in \mathscr{D}_1 \bigcup \mathscr{D}_2$ denotes the pairwise comparison $RT_{m,1} > RT_{m,2}$ or $RT_{m,1} \approx RT_{m,2}$. And the pairwise brain dynamics preference $(x_{n,m}^1, x_{n,m}^2)$ denotes the features recorded within the $n$-th channel for each preference proposition $\rho_m \forall m = 1, 2, \cdots, M_1 + M_2$.

In this paper, the 10s baseline (Fig. 2B) as the feature vector has been adopted, which is assumed to be long enough to detect any significant changes in brain activity (Zhang, 2000). This followed by exploring the relation between the 10s baseline $x$ ($\in R^k$) and the preference proposition $\rho_m$ under the following four assumptions: (1) different participants are independent during the data collection process; (2) Different EEG sensors used for recording are recorded independently from scalp without influencing other sensors (Homan et al., 1987; Teplan et al., 2002); (3) Different trials are conducted independently during the data collection process; (4) The collected response time are slightly corrupted by inherent (basically irremovable) sources of noise, but the ranking relationships are preserved to some extents.

**b) Brain Dynamics Preferences:** Again, instead of modelling the dependence between the RT and the 10s baseline as a regression problem, we aim to predict the orderings of the pairwise comparisons using the brain dynamics preferences, namely $f(x^1, x^2) \to \rho$. Further, a preference proposition $\rho$ has three states: $1, 0, -1$, denoting win ($RT_1 > RT_2$), tie ($RT_1 \approx RT_2$) and loss ($RT_1 < RT_2$), respectively.

However, classical classification models, e.g. logistic ordinal regression (Harrell, 2001), are infeasible for our problem, due to the lack of a normalized probability definition for three states. Since two types of preference propositions $\mathscr{D}_1$ and $\mathscr{D}_2$ are considered in our problem, we tailor-define a normalized probability definition, namely first normalizing the probabilities of the states $(1, -1)$

---

[1]We used the term "preference" intentionally to show that brain dynamics keep changing w.r.t. human behaviours and it happens because the human brain prefers one decision over others. Therefore, we prefer to call it"preference" than "classification".

(exclusively to $\mathscr{D}_1$) to 1, then generalize the probability definition to state 0. To be specific, it can be mathematically formulated as follows,

$$P(\rho|w,x^1,x^2) = \begin{cases} \sigma(w^T\Delta x)[1-\kappa(w^T\Delta x)] & \rho = 1 \\ \kappa(w^T\Delta x) & \rho = 0 \\ \sigma(-w^T\Delta x)[1-\kappa(w^T\Delta x)] & \rho = -1 \end{cases} \tag{2}$$

where $\sigma(z) = 1/(1+e^{-z})$ is the sigmoid function and $\sigma(-z) = 1 - \sigma(z)$. The symbol $\Delta x$ denotes the subtraction $(x^1 - x^2)$ between the brain dynamics preferences $(x^1, x^2)$. Following Weng & Lin (2011), the probability of a tie is modelled as the geometric mean between a win and a loss, namely $\kappa(w^T\Delta x) = \sqrt{\sigma(w^T\Delta x)[1-\sigma(w^T\Delta x)]}$.

**c) Preference State Transition:** Furthermore, considering the different functions of different regions in the human brain, the relative contributions of different channels to human response time may vary a lot. For example, the information conveyed by positive channels is positive related to the RT, while negative channels may convey the information which is negative related to the RT. There are also some noisy (non-relevant) channels which are independent to the learning task. Therefore, if we directly model the EEG preferences recorded in each channel without making any distinctions about the channel state (i.e. positive, noisy and negative), the model's reliability would inevitably degrade. Note that a channel is called as "noise" if the current algorithms could not extract useful brain information with EEG signals from this channel (Alharbi, 2018; Lin et al., 2018).

In the following, a transition matrix $\Pi_n$ is introduced to characterize the reliability of each channel $n$ w.r.t the corresponding task. Let $\rho$ denote the preference proposition and $\rho^{(n)}$ denote the prediction from the $n$-th channel. They are defined on the finite state space $S = \{1,0,-1\}$. Then we have

$$\Pi_n = P(\rho|\rho^{(n)}) = \begin{bmatrix} \pi_n & 0 & (1-\pi_n) \\ 0 & 1 & 0 \\ (1-\pi_n) & 0 & \pi_n \end{bmatrix}, \tag{3}$$

where $P_{i,j}(\rho|\rho^{(n)}) = P(\rho = S_j|\rho^{(n)} = S_i)$. Note that we do not consider the transition between the type-1 and type-2 preference propositions (i.e. $P(\rho = \{1,-1\}|\rho^{(n)} = 0) = 0$ and $P(\rho = 0|\rho^{(n)} = \{1,-1\}) = 0$), since the equal cases are hard to measure when do prediction. A promising approach to generalize the transition matrix $\Pi_n$ (Eq.3) is to introduce the concept of the confidence region to measure the equal cases (Pregibon et al., 1981).

*Remark:* The parameter $\pi_n$ in the transition matrix $\Pi_n$ actually indicates the reliability of the $n$-th channel $\forall n = 1, 2, \cdots, N$. It additionally helps to divide the channels into three states: (1) positive channels with $\pi_n$ close to 1, the ranking model (Eq.2) can extract enough information from the $n$-th channel, and exactly predict the state of the preference proposition. (2) Noisy channels with $\pi_n$ approximating to 0.5, the ranking model can not extract any useful information from the $n$-th channel. (3) Negative channels with $\pi_n$ close to 0, the ranking model can extract enough information from the $n$-th channel, but the prediction states are exactly opposite to the proposition states. The identified positive and negative channels are all considered as informative EEG channels, which helps in learning reliable models for the corresponding task. ∎

With the introduced transition matrix $\Pi_n$, the marginal likelihood function for each proposition $\rho$ can be further represented as $P(\rho) = \mathbb{E}_{\rho^{(n)}}\left[P(\rho|\rho^{(n)})P(\rho^{(n)})\right]$. Specifically,

$$P(\rho|w,\Pi_n,x^1,x^2) = \begin{cases} [\pi_n\sigma(w^T\Delta x) + (1-\pi_n)\sigma(-w^T\Delta x)][1-\kappa(w^T\Delta x_{n,m})] & \rho = 1 \\ \kappa(w^T\Delta x_{n,m}) & \rho = 0 \\ [(1-\pi_n)\sigma(w^T\Delta x) + \pi_n\sigma(-w^T\Delta x)][1-\kappa(w^T\Delta x_{n,m})] & \rho = -1 \end{cases} \tag{4}$$

For sake of simplicity, the subscripts $(n,m)$ are omitted. Let $\mathscr{D} = \mathscr{D}_1 \bigcup \mathscr{D}_2$ denotes all the preference propositions and $X$ represents the recorded EEG signals from $N$ different channels. We further extend our robust ordinal classification (Eq.4) for BDrank into the Bayesian version. A Gaussian prior is introduced for $w$ (i.e., $w \sim N(\mu, \Sigma)$). Since the transition matrix $\Pi_n$ only depends on the parameter $\pi_n$, we only focus on estimate the parameter $\pi_n$ $\forall n = 1, 2, \cdots, N$ in the following. Let $\pi$ denote $\{\pi_n\}_{n=1}^N$, and we introduce a Beta prior for each $\pi_n$ (i.e., $\pi \sim B(\alpha, \beta) = \prod_{n=1}^N B(\alpha_n, \beta_n)$). Then,

our BDrank can be formulated into a maximum a posteriori (MAP) estimate as follows,

$$\arg\max_{\pi,w} P_0(\pi)P_0(w)P(\mathscr{D}|\pi,w,X) = P_0(\pi)P_0(w)P(\mathscr{D}_1|\pi,w,X)P(\mathscr{D}_2|\pi,w,X)$$

$$= P_0(\pi)P_0(w)\prod_{n=1}^{N}\Big[\prod_{m=1}^{M_1}P(\rho_m=1|\pi_n,w,\Delta x_{n,m})\prod_{m'=1}^{M_2}P(\rho_{m'}=0|\pi_n,w,\Delta x_{n,m'})\Big] \tag{5}$$

$$= B(\pi|\alpha,\beta)N(w|\mu,\Sigma)\prod_{n=1}^{N}\Big[\prod_{m=1}^{M_1}\big([\pi_n\sigma(w^T\Delta x)+(1-\pi_n)\sigma(-w^T\Delta x)][1-\kappa(w^T\Delta x_{n,m})]\big)\prod_{m'=1}^{M_2}\kappa(w^T\Delta x_{n,m'})\Big].$$

Note that due to the symmetry of the state probability (Eq.2) and transition matrix (Eq.3) w.r.t. state 1 and $-1$, the resultant marginal likelihood (Eq.4) remains symmetry w.r.t. state 1 and $-1$. For simplicity, $\mathscr{D}_1$ is constructed using the preference propositions with state $\rho = 1$ only. $M_1$ and $M_2$ denote the number of type-1 and type-2 preference propositions, namely $|\mathscr{D}_1| = M_1$ and $|\mathscr{D}_2| = M_2$. The variable $n$ iterates over the channels. $m$ and $m'$ iterate over two types of preference propositions, respectively. Now our aim is to estimate the model parameters ($w$ and $\pi$) by maximizing Eq.5. In principle, any solution strategies for MAP can be considered to solve this problem. See Section 3 for optimization details.

**d) Reliability analysis and channel state estimation**   Note that our BDrank (Eq.5) indeed trains a mixture of two complementary classifiers, which share the same parameter $w$. It is different from classical mixture models, since it clusters at the channel level instead of the sample level. In particular, in terms of the positive channels with $\pi_n$ close to 1, BDrank relies on the first classifier to update the shared parameter $w$. In terms of the negative channels with $\pi_n$ close to 0, Eq.5 automatically switches to the opposite classifier which can extract correct information from the negative channels and update the shared parameter $w$ accordingly. Further, BDrank is robust to the noisy channels with $\pi_n$ approximating to 0.5, because Eq.5 gives up extracting information from the noisy channels by assigning a constant likelihood (i.e. 0.5) to each brain dynamics preference. Note that the estimated $\pi_n$ can be leveraged as the indicator to detect noisy channels with $\pi_n \approx 0.5, \forall n = 1, 2, \cdots, N$. See Fig. 4 in the experimental section for more details.

## 3   OPTIMIZATION METHODS

In this section, we describe a generalized Expectation-Maximization (EM) algorithm (Dempster et al., 1977) to solve the proposed BDrank (Eq.5). Since the feasible region of $\pi_n$ is restricted to $[0,1]$, the gradient-based optimization methods would make our solution inaccurate and inefficient. The EM algorithm is an efficient iterative procedure to compute the MAP problem in presence of latent variable ($\rho_m^{(n)}$ in Eq.5). EM avoids calculating the derivative to the expectation of the latent variable directly, and resorts to a surrogate lower bound to optimize. Therefore, EM, a silver bullet for MAP with latent variable, can significantly simplify the optimization over parameter $\pi_n$ for Eq.5.

### 3.1   GENERALIZED EM FOR BDRANK

For each type-1 proposition $\rho_m$, we introduce an auxiliary variable $\delta_m^{(n)}(\in \{1,0\})$ for each channel, representing the consistency between the preference proposition $\rho_m$ and the prediction $\rho_m^{(n)}$ given by the $n$-th channel. Specifically, $\delta_m^{(n)} = 1$ denotes the prediction $\rho_m^{(n)}$ given by the first classifier is consistent with the preference proposition $\rho_m$, and $\delta_m^{(n)} = 0$ denotes the prediction $\rho_m^{(n)}$ estimated by the second classifier is consistent with the preference proposition $\rho_m$. We can therefore find an equivalent formulation of our BDrank model involving the auxiliary variable $\Xi = \{\delta_m^{(n)}\}_{m=1}^{M_1}$.

$$P(\mathscr{D},\Xi,\pi,w|X) = P_0(\pi)P_0(w)P(\mathscr{D}_1,\Xi|\pi,w,X)P(\mathscr{D}_2|w,\pi,X)$$

$$= P_0(\pi)P_0(w)\prod_{n=1}^{N}\Big[\prod_{m=1}^{M_1}P(\rho_m=1,\delta_m^{(n)}|\pi_n,w,\Delta x_{n,m})\prod_{m'=1}^{M_2}P(\rho_{m'}=0|\pi_n,w,\Delta x_{n,m'})\Big] \tag{6}$$

$$= P_0(\pi)P_0(w)\prod_{n=1}^{N}\Big[\prod_{m=1}^{M_1}[\pi_n\sigma(w^T\Delta x_{n,m})]^{\delta_m^{(n)}}\big[(1-\pi_n)\sigma(-w^T\Delta x_{n,m})\big]^{1-\delta_m^{(n)}}[1-\kappa(w^T\Delta x_{n,m})]\prod_{m'=1}^{M_2}\kappa(w^T\Delta x_{n,m'})\Big].$$

Now, we can deal with the joint distribution directly, which leads to significant simplifications for optimization. The complete log likelihood can be written as

$$\log P(\mathscr{D}, \Xi, \pi, w | X) = \log P_0(\pi) + \log P_0(w) + \sum_{n=1}^{N} \sum_{m=1}^{M_1} \log[1 - \kappa(w^T \Delta x_{n,m})] + \sum_{n=1}^{N} \sum_{m'=1}^{M_2} \log \kappa(w^T \Delta x_{n,m'}) \quad (7)$$

$$+ \sum_{n=1}^{N} \sum_{m=1}^{M_1} \left[ \delta_m^{(n)} \log \pi_n \sigma(w^T \Delta x_{n,m}) + (1 - \delta_m^{(n)}) \log(1 - \pi_n) \sigma(-w^T \Delta x_{n,m}) \right].$$

*Expectation Step* In the expectation step, we first calculate the expected value of the auxiliary variable $\delta_m^{(n)}$ w.r.t its posterior distribution $P(\delta_m^{(n)} | \pi, w, \rho_m = 1, x_{n,m}) \; \forall n = 1, 2, \cdots, N, \forall m = 1, 2, \cdots, M_1$:

$$E_{P(\delta_m^{(n)} | \pi, w, \rho_m = 1, x_{n,m})}[\delta_m^{(n)}] = \frac{P(\rho_m = 1, \delta_m^{(n)} | \pi_n, w, \Delta x_{n,m})}{P(\rho_m = 1 | \pi_n, w, \Delta x_{n,m})} = \frac{1}{1 + \frac{(1 - \pi_n) \sigma(-w^T \Delta x_{n,m})[1 - \kappa(w^T \Delta x_{n,m})]}{\pi_n \sigma(w^T \Delta x_{n,m})[1 - \kappa(w^T \Delta x_{n,m})]}} \triangleq \gamma(\delta_m^{(n)}),$$
$$(8)$$

where $\gamma(\delta_m^{(n)})$ denotes the degree of the consistency between the prediction $\rho_m^{(n)}$ and the preference proposition $\rho_m$. Then, the expected value of the complete-data log likelihood function w.r.t the posterior expectation of the auxiliary variable $\Xi = \{\delta_m^{(n)}\}_{m=1}^{M_1}$ can be represented as follows:

$$\mathscr{L}(w, \pi) = \sum_{n=1}^{N} \left[ (\alpha_n - 1) \log \pi_n + (\beta_n - 1) \log(1 - \pi_n) \right] - \frac{1}{2} (w - \mu)^T \Sigma^{-1} (w - \mu) + \sum_{n=1}^{N} \sum_{m=1}^{M_1} \log[1 - \kappa(w^T \Delta x_{n,m})]$$

$$+ \sum_{n=1}^{N} \sum_{m'=1}^{M_2} \log \kappa(w^T \Delta x_{n,m'}) + \sum_{n=1}^{N} \sum_{m=1}^{M_1} \left[ \gamma(\delta_m^{(n)}) \log \pi_n + (1 - \gamma(\delta_m^{(n)})) \log(1 - \pi_n) \right] \quad (9)$$

$$+ \sum_{n=1}^{N} \sum_{m=1}^{M_1} \left[ \gamma(\delta_m^{(n)}) \log \sigma(w^T \Delta x_{n,m}) + (1 - \gamma(\delta_m^{(n)})) \log \sigma(-w^T \Delta x_{n,m}) \right] + constant.$$

*Maximization Step* In the maximization step, we maximize the objective function Eq.9 w.r.t the model parameters $\pi$ and $w$, respectively. In terms of $\pi$, we set the gradient of Eq.9 w.r.t $\pi_n$ to zero and obtain the following estimate for $\pi_n$:

$$\pi_n^{new} = \frac{\sum_{m=1}^{M_1} \gamma(\delta_m^{(n)}) + \alpha_n - 1}{M_1 + \alpha_n + \beta_n - 2}, \qquad n = 1, 2, \cdots, N. \quad (10)$$

In terms of $w$, due to the complexity of the sigmoid function, we do not have a closed form solution for $w$ and we need to use gradient-based optimization methods. In the following, we adopt the L-BFGS to optimize $w$, the objective function $\mathscr{L}(w)$ consists of the second, the third, the fourth and the sixth parts of Eq.9 and the gradient function $g(w)$ can be represented as follows,

$$g(w) = -\Sigma^{-1}(w - \mu) + \sum_{n=1}^{N} \sum_{m=1}^{M_1} \frac{1 - 2\sigma(w^T \Delta x_{n,m})}{2(1 - \frac{1}{\kappa(w^T \Delta x_{n,m})})} \Delta x_{n,m} + \sum_{n=1}^{N} \sum_{m=1}^{M_1} (\gamma(\delta_m^{(n)}) - \sigma(w^T \Delta x_{n,m})) \Delta x_{n,m}$$

$$+ \frac{1}{2} \sum_{n=1}^{N} \sum_{m'=1}^{M_2} (1 - 2\sigma(w^T \Delta x_{n,m'})) \Delta x_{n,m'}. \quad (11)$$

$w^{new}$ can be solved with L-BFGS using $\mathscr{L}(w)$ and $g(w)$. The EM algorithm then iterates the E-step and M-step until convergence is achieved.

## 3.2 ONLINE GEM FOR BDRANK

The generalized EM approach introduced in Section 3.1 is inefficient for large-scale datasets, because we need to iteratively calculate the gradient with respect to parameters $\pi$ and $w$ over all samples during each maximization step. Motivated from the stochastic approximation literature, we introduce an online-generalized Expectation Maximization (OnlineGEM) approach, which resorts to stochastic mini-batch optimisation to learn the parameters. To be specific, OnlineGEM approximates the updated $\pi$ and $w$ in batch EM with a single sample or mini-batch samples. Since a mini-batch samples cannot be a perfect approximation for the whole dataset, we interpolate between the new and former estimations with a decreasing step-size $\eta_k{}^2$, as in Liang & Klein (2009).

---

[2] $\eta_k = (k+2)^{-\tau_0}$, where $k$ is the number of iterations and $0.5 < \tau_0 < 1$. The smaller the $\tau_0$, the larger the update $\eta_k$, and the more quickly we forget (decay) our old parameters. This can lead to swift progress but also generates instability.

Before the $k$-th iteration, we randomly downsample a mini-batch $\mathscr{D}^k$ from the preference propositions $\mathscr{D}$. The number of two types preference propositions in $\mathscr{D}^k$ are $M_1^*$ and $M_2^*$, which are much smaller than the corresponding total size $M_1$ and $M_2$, respectively.

The expectation step remains similar. The difference is that we only need to calculate the posterior expectation of the auxiliary variable $\delta_m^{(n)}$ over the mini-batch $\mathscr{D}^k$.

In the maximization step, we maximize the objective function, calculated on the mini-batch $\mathscr{D}^k$, with regard to model parameters $\pi$ and $w$. In terms of parameter $\pi_n$, since its marginal distribution belongs to exponential family, we could perform the stochastic update in the space of sufficient statistics (Cappé & Moulines, 2009). $\phi_n^*$ denotes the noisy estimate of the sufficient statistic for $\pi_n$.

$$\phi_n^* = \frac{M_1}{M_1^*} \sum_{m \in \mathscr{D}^k} \gamma(\delta_m^{(n)}), \tag{12a}$$

$$\phi_n^{(k)} = (1 - \eta_k)\phi_n^{(k-1)} + \eta_k\phi_n^*, \tag{12b}$$

$$\pi_n^{new} = \frac{\phi_n^{(k)} + \alpha_n - 1}{M_1 + \alpha_n + \beta_n - 2}, \qquad n = 1, 2, \cdots, N. \tag{12c}$$

In terms of parameter $w$, the above practice is infeasible due to its non-exponential marginal distribution. Inspired by the stochastic gradient EM algorithms in Cappé & Moulines (2009), we perform the stochastic update in the original space. First, a local optima regression weight $w^{(k)}$ can be estimated via iterative optimization over the mini-batch $\mathscr{D}^k$, using L-BFGS algorithm. Then we interpolate between the local optima and former estimations to form a global approximation.

$$w^{(k)} = \text{L-BFGS}(\mathscr{L}(w), g(w), \mathscr{D}^k) \tag{13a}$$

$$w^{new} = (1 - \eta_k)w^{old} + \eta_k w^{(k)}. \tag{13b}$$

The convergence issues of the proposed online generalized EM algorithm are the analogues of the discussion given by Cappé & Moulines (2009) for their stochastic gradient EM Algorithms. The existence of such links is hardly surprising. In view of the discussions in Section 3 of Cappé & Moulines (2009), the online update rule (Eq.13b) could also be seen as a stochastic gradient descent formula, namely $w^{new} = w^{old} + \eta_k(w^{(k)} - w^{old})$.

## 4 EMPIRICAL ANALYSIS

In this section, we demonstrate the reliability of the proposed BDrank (Eq.5) with EEG signals from forty four participants.

**Data Preprocessing:** EEG preferences for each participant has been generated as follows: (1) the trials of each participant were randomly divides into two parts: 50% for training and 50% for test; (2) EEG preferences were constructed according to the pairwise comparisons between the RTs. To be specific, the type-1 preference propositions $\mathscr{D}_1$ were constructed with RT comparisons $(RT_{m,1}, RT_{m,2})$, which satisfies $RT_{m,2} < \min(RT_{m,2} + \tau_1, \tau_2 \times RT_{m,2}) < RT_{m,1}$; the type-2 preference propositions $\mathscr{D}_2$ were constructed with RT comparisons $(RT_{m',1}, RT_{m',2})$, which satisfies $RT_{m',2} < RT_{m',1} < \min(RT_{m',3} + \tau_3, \tau_4 \times RT_{m',2})$. It is notable that $\tau_1 > \tau_3 > 0$ and $\tau_2 > \tau_4 > 1$ control the difference in the RT comparisons simultaneously, we empirically set $\tau_1 = 1; \tau_2 = 1.5; \tau_3 = 0.8; \tau_4 = 1.3$ for all participants in our experiment setting. Considering the time delay among the channels in the time domain, Fourier transform (Welch, 1967) has been applied to EEG signals to transform time-series into frequency domain. Further, to avoid overhead computation, EEG power within 0-30Hz has been selected, which is considered to be the most relevant to the RTs (Huang et al., 2015).

**Baselines and Metrics:** We compared BDrank with one regression model Support Vector Regression (SVR) (Smola & Schölkopf, 2004) and one ordinal classification model Logistics Ordinal Regression (LOR) (Harrell, 2001). First, we aggregate the predictions from different channels using a simple voting scheme, namely $\hat{\rho}_m = sign\left(\sum_{n=1}^N \rho_m^{(n)}\left[\mathbf{1}(\pi_n > \kappa) - \mathbf{1}(\pi_n < 1 - \kappa)\right]\right)$. $\rho_m^{(n)}$ denotes the predicted state (1 means win and -1 means loss) for $(RT_m^1, RT_m^2)$ by the $n$-th channel, using the brain dynamics preference $(x_{n,m}^1, x_{n,m}^2)$. $\hat{\rho}_m$ is the final estimated order for $(RT_m^1, RT_m^2)$ by aggregating the predictions $\rho_m^{(n)}$ over all channels. $\mathbf{1}()$ is an indicator that returns one if the argument is valid and returns zero otherwise.

Then, we introduce two metrics to measure the performance of BDrank model from different perspectives. First, we adapted the Wilcoxon-Mann-Whitney statistics (Yan et al., 2003) to evaluate the accuracy (in %, higher is better) over all pairs, namely $\frac{1}{M_1}\sum_{m=1}^{M_1}\mathbf{1}(\rho_m = \hat{\rho}_m)$. Further, we investigate the reliability of BDrank in terms of preserving the global ordering w.r.t RTs. Note that a totally ordered set could be equally represented by a fully directed graph, where the fully directed graph can be further encoded by its degree sequence. Therefore, we first collected the indegree sequences[3] (Becirovic, 2017) of the constructed directed graph using the predicted RTs and then measured the discrepancy between the predicted indegree sequences and ground truth using the root-mean-squared errors (smaller is better). See Supplementary for the detailed description.

Note that we only trust the predictions from informative channels with reliability $\pi_n > \kappa$ or $\pi_n < 1 - \kappa$. $\kappa$ is set to 0.85 for all participants in our experiment. In terms of SVR and LOR, consider the scale difference between the EEG signals between different channels, we train a SVR/LOR model for each channel and aggregate the results from different channels to calculate the final predictions using the majority voting scheme. Since there is no mechanism for SVR and LOR to evaluate the channel state, we trust all the channels by default. Further, we only calculate the two metrics on the type-1 preference propositions $\mathscr{D}_1$, since the state of the type-2 preference propositions $\mathscr{D}_2$ is hard to evaluate when do prediction.

**Parameter Initialization:** In terms of the weight $w$, we randomly initialized $w$ in $[-1, 1]$. In terms of the channel reliability $\pi_n$, we assumed all channels were relevant to the task beforehand, and randomly initialized the channel reliability $\pi_n, \forall n = 1, 2, \cdots, N$ in $[0.5, 1]$. The L-BFGS implementation was downloaded from Granzow, where we used the default parameters. In terms of the hyperparameters $(\mu, \Sigma)$ for $w$, we adopted the standard Gaussian distribution, namely $w \sim N(\mathbf{0}, \mathbf{1})$. In terms of the hyperparameters $(\alpha_n, \beta_n)$, as we intended to eliminate the effects of noisy channels, we adopted a strong non-informative prior for $\pi_n$, namely $\alpha_n = \beta_n = 20, \forall n = 1, 2, \cdots, N$, according to Bishop (2006). In terms of the maximum iteration number, we set $MaxIter = 30$ in our experiment to ensure the algorithm converged for each participant. In terms of the minibatch size $R$, we set the sample ratio to 10% and 5% for type-1 and type-2 preference propositions, respectively. The parameters for LOR were optimized with L-BFGS with the default setting for all participants. In terms of SVR, we resorted to 5-fold cross validation to find the best parameters for each participant.

### 4.1 EMPIRICAL RESULTS OF BDRANK ON BRAIN DYNAMICS PREFERENCES

The accuracies of LOR and BDrank for training and test EEG preferences are presented in Table 1. For SVR, we followed standard procedure and optimization for each participant to predict RTs with training and test EEG signals. We adopted the Wilcoxon-Mann-Whitney statistics to measure the accuracy of the predicted RTs w.r.t. ground truth (shown in Table 1).

Table 1: Training and test accuracy (in %). Higher is better, the best is marked in gray

| Participant | | P1 | P2 | P3 | P4 | P5 | P6 | P7 | P8 | P9 | P10 | P11 | P12 | P13 | P14 | P15 | P16 | P17 | P18 | P19 | P20 | P21 | P22 |
|---|---|---|---|---|---|---|---|---|---|---|---|---|---|---|---|---|---|---|---|---|---|---|---|
| Train | SVR | 98.16 | 98.45 | 100 | 99.40 | 100 | 100 | 100 | 100 | 100 | 100 | 100 | 98.31 | 98.73 | 99.82 | 99.69 | 100 | 100 | 100 | 100 | 98.78 | 99.64 | 100 |
| | LOR | 91.66 | 94.73 | 77.24 | 88.65 | 81.31 | 97.10 | 96.37 | 85.42 | 84.35 | 73.67 | 82.42 | 89.97 | 89.79 | 85.19 | 91.83 | 89.01 | 88.38 | 86.06 | 91.90 | 91.65 | 88.66 | 84.83 |
| | BDrank | 98.96 | 97.13 | 95.26 | 99.08 | 94.92 | 99.85 | 97.45 | 97.40 | 97.33 | 93.60 | 99.43 | 97.99 | 98.48 | 96.47 | 98.60 | 99.88 | 98.95 | 100.00 | 99.68 | 99.03 | 96.23 | 85.95 |
| Test | SVR | 68.56 | 76.81 | 68.49 | 61.20 | 76.04 | 78.04 | 69.59 | 66.90 | 67.01 | 79.23 | 73.34 | 67.58 | 69.45 | 69.47 | 85.38 | 73.73 | 75.38 | 55.60 | 66.55 | 71.87 | 75.03 | 68.22 |
| | LOR | 71.22 | 69.12 | 53.12 | 61.63 | 64.63 | 68.46 | 70.99 | 54.97 | 67.18 | 59.18 | 52.96 | 69.87 | 70.50 | 61.18 | 76.27 | 69.52 | 50.35 | 60.87 | 82.73 | 60.67 | 65.10 | 69.98 |
| | BDrank | 77.04 | 77.93 | 84.05 | 71.93 | 81.21 | 68.06 | 73.45 | 79.64 | 72.97 | 82.09 | 68.92 | 76.24 | 72.95 | 67.87 | 87.64 | 73.74 | 63.08 | 65.53 | 83.85 | 74.62 | 74.38 | 77.28 |

| Participant | | P23 | P24 | P25 | P26 | P27 | P28 | P29 | P30 | P31 | P32 | P33 | P34 | P35 | P36 | P37 | P38 | P39 | P40 | P41 | P42 | P43 | P44 |
|---|---|---|---|---|---|---|---|---|---|---|---|---|---|---|---|---|---|---|---|---|---|---|---|
| Train | SVR | 100 | 100 | 97.13 | 100 | 99.47 | 100 | 99.88 | 99.98 | 100 | 100 | 100 | 100 | 99.93 | 99.97 | 99.95 | 100 | 98.21 | 99.09 | 100 | 100 | 100 | 99.59 |
| | LOR | 91.63 | 98.08 | 88.86 | 92.10 | 90.97 | 84.00 | 86.91 | 92.00 | 87.36 | 90.84 | 80.20 | 86.10 | 91.40 | 89.03 | 93.47 | 93.52 | 90.86 | 92.19 | 93.42 | 82.37 | 81.87 | 81.80 |
| | BDrank | 97.13 | 99.79 | 94.00 | 99.19 | 95.90 | 92.29 | 98.59 | 96.60 | 94.64 | 92.61 | 85.70 | 92.32 | 99.09 | 96.37 | 98.64 | 99.21 | 97.89 | 95.94 | 96.74 | 93.41 | 96.20 | 76.04 |
| Test | SVR | 69.70 | 72.84 | 67.25 | 80.89 | 72.71 | 66.17 | 77.24 | 76.59 | 81.11 | 79.54 | 80.64 | 73.83 | 78.39 | 70.99 | 75.34 | 61.9 | 86.92 | 76.51 | 69.89 | 68.21 | 65.09 | 63.06 |
| | LOR | 69.80 | 74.24 | 72.02 | 63.80 | 73.62 | 66.46 | 63.05 | 77.12 | 73.09 | 72.15 | 73.20 | 69.21 | 69.53 | 71.31 | 70.01 | 84.42 | 69.85 | 65.87 | 70.02 | 77.20 | 84.89 | 70.41 |
| | BDrank | 71.40 | 82.06 | 74.06 | 72.61 | 79.12 | 68.56 | 81.69 | 79.99 | 75.57 | 77.67 | 79.34 | 79.30 | 80.11 | 73.97 | 76.14 | 86.93 | 87.23 | 77.47 | 70.40 | 77.38 | 88.47 | 74.16 |

From Table 1, we observe that: (1) SVR achieves the highest accuracies (100% accuracy) on the training EEG data for almost all participants. However, test accuracies for most participants are inferior to other models, 21 participants for LOR and 35 participants for BDrank out of 44 participants. This observation is consistent with our previous discussion that regression-based models are easily overfitting to the training datasets, especially when extreme values (RTs in our problem) exist. (2) BDrank shows significant improvements over SVR and LOR in terms of test accuracies.

---

[3]We only consider the indegree sequence because the indegree and outdegree of a vertex can be uniquely determined when the overall degree of each vertex is fixed. The indegree of vertex $v_t$ can be calculated as $\deg^-(v_t) = \sum_{m \in N_1(v_t)}[\mathbf{1}(\hat{\rho}_m = 1) + 0.5 \times \mathbf{1}(\hat{\rho}_m = 0)] + \sum_{m \in N_2(v_t)}[\mathbf{1}(\hat{\rho}_m = -1) + 0.5 \times \mathbf{1}(\hat{\rho}_m = 0)]$, where $N_1(v_t), N_2(v_t)$ denote the index set of the pairwise comparisons with the RT of trial $t$ (vertex $v_t$) appearing in the first and second position, respectively.

In particular, the number of participants with test accuracy above 75% are 17 and 25 for SVR and BDrank, respectively. This observation is consistent with our statement that classification-based models, served as a regularized alternative for the regression-based models, can effectively circumvent the overfitting caused by non-smooth RTs and preserve the ordering corresponding to RTs. (3) However, simple classification-based extensions (e.g. LOR) do not enjoy the benefits, which only achieves comparable test accuracies with SVR (outperforming SVR on only 21/44 participants). It is because they are not robust to the noisy data, while it is a common phenomenon when dealing with EEG signals. (4) Note that, compared to the fine-tuned SVR for each participant, fixed parameters were adopted for all participants in the experiment of LOR and BDrank. It further verifies the superior robustness of ordinal classification models compared to regression-based models.

To further investigate the reliability of BDrank in terms of preserving the global ordering corresponding to RTs, we first collected the indegree sequences of the constructed directed graph using the predicted RTs and then measured the indegree discrepancy between the calculated indegree sequences and the ground truth using the root-mean-squared error (RMSE) (shown in Table. 2).

Table 2: Training and test RMSE (in #). Smaller is better, the best is marked in gray

| Participant | | P1 | P2 | P3 | P4 | P5 | P6 | P7 | P8 | P9 | P10 | P11 | P12 | P13 | P14 | P15 | P16 | P17 | P18 | P19 | P20 | P21 | P22 |
|---|---|---|---|---|---|---|---|---|---|---|---|---|---|---|---|---|---|---|---|---|---|---|---|
| Train | SVR | 1.52 | 2.51 | 0.00 | 1.23 | 0.00 | 0.00 | 0.00 | 0.00 | 0.00 | 0.00 | 0.00 | 2.01 | 1.26 | 0.39 | 0.64 | 0.00 | 0.00 | 0.00 | 0.00 | 3.84 | 1.56 | 0.00 |
| | LOR | 4.06 | 4.69 | 16.60 | 9.54 | 29.77 | 3.31 | 5.30 | 13.79 | 28.86 | 42.19 | 16.04 | 6.38 | 5.53 | 11.66 | 8.42 | 1.25 | 8.12 | 4.90 | 8.63 | 15.02 | 20.88 | 31.94 |
| | BDrank | 0.92 | 3.78 | 3.16 | 0.74 | 7.52 | 0.11 | 4.67 | 1.82 | 3.85 | 11.38 | 0.65 | 1.73 | 0.91 | 3.18 | 1.71 | 0.39 | 0.52 | 0.00 | 2.91 | 2.15 | 7.51 | 34.12 |
| Test | SVR | 14.30 | 19.52 | 22.95 | 30.11 | 39.45 | 16.80 | 29.69 | 27.01 | 53.23 | 38.90 | 23.56 | 20.32 | 13.71 | 21.55 | 14.42 | 26.66 | 16.40 | 16.24 | 28.40 | 47.88 | 39.85 | 60.94 |
| | LOR | 13.74 | 23.08 | 31.36 | 27.96 | 53.02 | 22.61 | 33.03 | 34.91 | 46.77 | 62.14 | 35.40 | 20.02 | 13.71 | 26.61 | 23.03 | 30.98 | 28.80 | 15.43 | 24.52 | 62.99 | 56.10 | 59.79 |
| | BDrank | 10.94 | 19.45 | 10.68 | 19.37 | 25.12 | 10.60 | 28.59 | 12.31 | 32.92 | 27.63 | 16.13 | 14.82 | 12.67 | 21.75 | 11.54 | 26.32 | 12.95 | 3.35 | 19.38 | 32.09 | 40.92 | 52.11 |

| Participant | | P23 | P24 | P25 | P26 | P27 | P28 | P29 | P30 | P31 | P32 | P33 | P34 | P35 | P36 | P37 | P38 | P39 | P40 | P41 | P42 | P43 | P44 |
|---|---|---|---|---|---|---|---|---|---|---|---|---|---|---|---|---|---|---|---|---|---|---|---|
| Train | SVR | 0.00 | 0.00 | 6.08 | 0.00 | 2.22 | 0.00 | 0.98 | 0.22 | 0.00 | 0.00 | 0.00 | 0.26 | 0.19 | 0.26 | 0.00 | 2.10 | 1.38 | 0.00 | 0.00 | 0.00 | 0.00 | 0.67 |
| | LOR | 16.60 | 3.31 | 15.98 | 19.20 | 17.59 | 27.85 | 22.97 | 14.68 | 28.61 | 17.74 | 36.52 | 23.72 | 7.71 | 11.56 | 7.46 | 6.22 | 6.05 | 5.88 | 11.70 | 22.95 | 15.26 | 11.39 |
| | BDrank | 5.05 | 0.45 | 7.70 | 1.94 | 9.65 | 13.36 | 2.79 | 6.21 | 10.07 | 16.75 | 28.45 | 13.62 | 1.02 | 4.07 | 1.20 | 1.18 | 1.81 | 3.39 | 6.58 | 22.79 | 12.28 | 15.43 |
| Test | SVR | 42.89 | 25.60 | 40.55 | 41.12 | 44.93 | 49.09 | 39.85 | 36.77 | 39.03 | 37.63 | 35.72 | 39.49 | 16.90 | 24.19 | 21.14 | 15.44 | 9.15 | 21.64 | 46.41 | 35.09 | 26.92 | 20.37 |
| | LOR | 46.80 | 22.47 | 36.34 | 64.61 | 41.59 | 48.48 | 31.45 | 40.34 | 53.32 | 48.30 | 47.65 | 35.17 | 22.46 | 26.83 | 25.55 | 13.59 | 18.60 | 22.05 | 47.83 | 26.00 | 20.95 | 18.10 |
| | BDrank | 44.58 | 14.89 | 32.64 | 32.46 | 37.91 | 39.16 | 24.13 | 29.46 | 38.94 | 40.12 | 39.23 | 34.7 | 15.26 | 24.18 | 18.03 | 8.8 | 9.13 | 14.26 | 42.25 | 23.86 | 9.09 | 16.40 |

From Table. 2, we observe that: (1) On the training datasets, SVR could exactly recover the indegree sequence (RMSE equalling to 0), which means SVR preserves the global ordering corresponding to RTs; while LOR and BDrank could not fully fit the training datasets. (2) Note that, on the test datasets, BDrank shows surprisingly good generalization performance and maximum preserves the global ordering corresponding to RTs.

To further demonstrate the superiority of our BDrank, we calculated the indegree sequences using the predicted RTs for participants P19, P38, P42, P43 with the most representative performance (Fig. 3). Similar observations can be found for other participants.

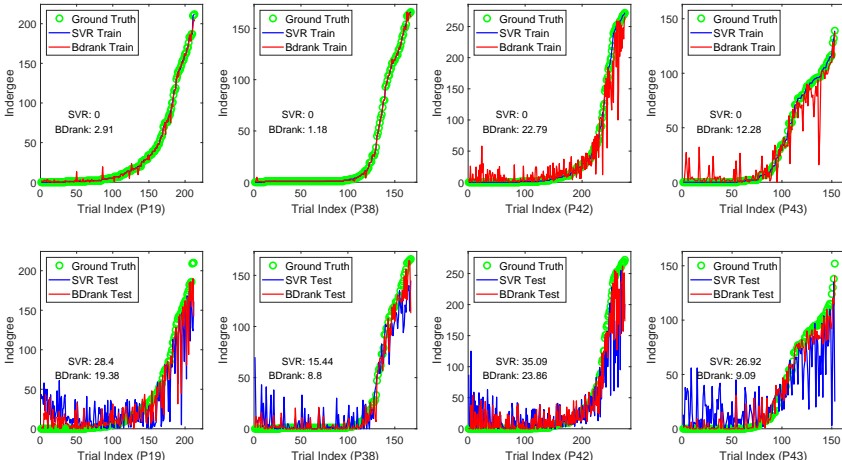

Figure 3: Indegree sequence for BDrank and SVR (closer is better). The root-mean-squared error (RMSE) was also measured between the estimated indegree sequences and the ground truth.

From Fig. 3, we observe that: (1) On the training datasets, the indegree sequences predicted by SVR completely overlaps with the ground truth; while the indegree sequence predicted by BDrank fluctuates locally but retains the overall trend. (2) On the test datasets, the indegree sequences predicted by BDrank still closely aligns with the ground truth with slight fluctuates; while the indegree sequences predicted by SVR fluctuates significantly and fails to maintain the trend with the ground truth, e.g. $P19, P43$. (3) It is interesting to note that the indegree sequences predicted by SVR usually fluctuates heavily for low indegree trials (denoting small RTs) and high indegree trials (denoting large

RTs). It means that SVR over-estimates the RTs with small values and under-estimates the RTs with large values. It is consistent with our claim that regression-based model is not suitable for the tasks with non-smooth response variable.

## 4.2 NOISY CHANNEL DETECTION

We also investigated the reliability of our BDrank from the perspective of noisy channel detection. According to our analysis, the parameter $\pi_n$ in the transition matrix $\Pi_n$ actually indicates the channel reliability. Hereafter, we leverage $\pi_n$ as the channel reliability indicator to detect noisy channels. Fig. 4 lists the noisy channels (marked in red) detected with $0.15 \leq \pi_n \leq 0.85$, $\forall n = 1, 2, \cdots, N$.

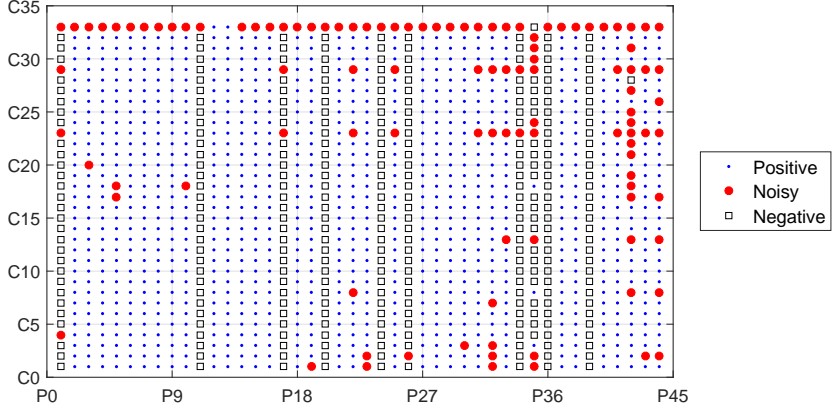

Figure 4: Reliability of different channels for forty four participants estimated by BDrank. Each column denotes the states of 33 channels for each participant. The channels with estimated reliability $0.15 \leq \pi_n \leq 0.85$ are marked in red.

Fig. 4 shows that: (1) the noisy channels is universally exist among the EEG signals. There are total 42 among 44 participants with at least one noisy channels detected. For example, the 33-th channel is recognized as the noisy channel by BDrank for almost all participants. It is reasonable since the 33-th channel is the non-EEG channel, which is generally acknowledged as the non-relevant channel to any tasks; (2) For each participant, most channels are reliable, which ensures we can always find enough support to training our BDrank; and (3) The detected noisy channels varies from participant to participant, and do not possess the transitivity property between participants. Because the noise can arise due to (i) intrinsic non-EEG channel, e.g. the 33-th channel; (ii) channels for lateral mastoid references, e.g. the 23-th and 29-th channel (Chatrian et al., 1985); and (iii) improper experimentation or artifacts (Lin et al., 2018).

## 5 CONCLUSION

This work proposes a BDrank model to assess the state of mental fatigue. The efficacy of BDrank model was demonstrated using EEG data collected in sustained driving task from 44 participants. This model has been further combined with an online-generalized Expectation Maximization (OnlineGEM) algorithm to provide a continuous update in the model. BDrank model utilized a unique methodology with a regularized alternative, i.e. ordinal classification, to circumvent overfitting to the extreme values of RTs. It has been demonstrated that the overall performance of BDrank can be significantly improved with the introduction of the transition matrix, which enables the technique to evaluate the reliability of informative EEG channels while detecting noisy EEG channels. Empirical results show that BDrank combined with the OnlineGEM algorithm delivers significant improvement over SVR and LOR in terms of global ranking preservation.

In this work, the cooperation mechanism among channels is simplified as a weighted voting system, while different trials are viewed independently. We intend to further formulate it with more complex mechanisms, such as Markov decision process (MDP), to conduct learning and decision making simultaneously. Some previous (Chen et al., 2016; 2015) studied the decision making process among crowd (noisy) workers, which is promising to our setting to investigate the cooperation mechanism among noisy channels. Efforts are underway to apply this approaches in future work.

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

SUPPLEMENTARY:

**The discrepancy between the predicted indegree sequence and the ground truth:**

Assume there exists a ranking list $S : a > b > c > d$, we can constructed 6 ($= \frac{4 \times 3}{2}$) preference propositions $\{\rho_i\}_{i=1}^6$ with the pairwise comparisons. Further, these could be equally represented by the fully directed graph with four vertices as Fig.5a. Then, the indegree of each vertex $v_i$ can be calculated following $\deg^-(v_i) = \sum_{m \in N_1(v_i)} \mathbf{1}(\rho_m = 1) + \sum_{m \in N_2(v_i)} \mathbf{1}(\rho_m = -1)$, where $N_1(v_i), N_2(v_i)$ denote the index set of the pairwise comparisons with the vertex $i$ appearing in the first and second position, respectively. Therefore, the indegree sequence (in ascending order) of Fig.5a can be represented as $X = [0, 1, 2, 3]$ for the corresponding vertex sequence $dcba$. Note that we only consider the indegree sequence because the indegree and outdegree of a vertex can be uniquely determined when the overall degree of each vertex is fixed.

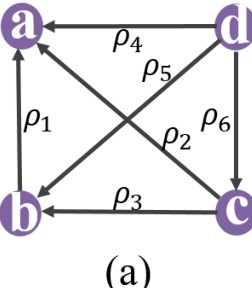 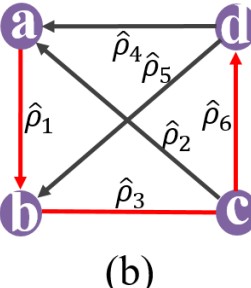

(a)           (b)

Figure 5: Directed graph representation of the ranking list $S : a > b > c > d$. **Left Panel**: the ground truth, where $\{\rho_i\}_{i=1}^6$ is the constructed preference proposition. **Right Panel**: the prediction, where $\{\hat{\rho}_i\}_{i=1}^6$ is the predicted order for each pairwise comparison. Note that we use the edges marked in red (e.g. $\{\hat{\rho}_1, \hat{\rho}_3, \hat{\rho}_6\}$) to denote the wrong predictions.

As for each prediction model, we first calculate the final estimation $\{\hat{\rho}_i\}_{i=1}^6$ for each pairwise comparisons using a simple voting scheme, detailed in Section 4. The corresponding fully directed graph is shown in Fig.5b. We calculate the indegree of each vertex $v_i$ following $\deg^-(v_i) = \sum_{m \in N_1(v_i)} [\mathbf{1}(\hat{\rho}_m = 1) + 0.5 \times \mathbf{1}(\hat{\rho}_m = 0)] + \sum_{m \in N_2(v_i)} [\mathbf{1}(\hat{\rho}_m = -1) + 0.5 \times \mathbf{1}(\hat{\rho}_m = 0)]$. Note that we consider the tie case (e.g. $\rho = 0$) here since the number of votes for state $1$ and $-1$ may be comparable. Next, the predicted indegree sequence for the vertex sequence $dcba$ is $\hat{X} = [1, 0.5, 2.5, 2]$. Therefore the discrepancy between the predicted indegree sequences and ground truth using the root-mean-squared errors (RMSD) can be calculated as follows,

$$\text{RMSD}(X) = \sqrt{\frac{\sum_{i=1}^T (X_i - \hat{X}_i)^2}{T}} = 0.7906.$$

Then, we can use this value to investigate the reliability of the proposed method in terms of preserving the global ordering.

