# OpenReview forum: "Mental Fatigue Monitoring using Brain Dynamics Preferences"
_ICLR.cc/2019/Conference_

### Official Review · AnonReviewer1 · 2018-10-20
**Bdrank for fatigue monitoring of EEG driving simulator experiments**

**Rating:** 2
**Confidence:** 5

**Review:**

The paper studies fatigue monitoring of EEG driving simulator experiments using various EEG analysis algorithms, one also based on ranking. The data used was from a prior experiment.

The paper is written in a rather confusing manner, which makes the assessment of originality and significance a hard task for the reviewer. A novel algorithm Bdrank (based on raking is defined) and compared to 2 other algorithms; unclear why with these and not with others. The paper ignores a large portion of the literature, starting with Kohlmorgen et al 2007, Blankertz group, Lee group etc.
The results  are only somewhat interesting, no understanding of the underlying physiological processes is given.

Overall, I consider the paper somewhat preliminary.

---

> ### Author Response · Authors · 2018-11-13
> **SVR and LOR are selected with careful consideration.**
>
> Thanks for your comments and questions regarding our paper. We summarize your comments into the following three sub-problems and answer each one separately.
>
> Q1: It is unclear why the paper compares with two algorithms (SVR and LOR) and not with others.
> A1: Considering the non-smooth property of human response time during data collection, regular regression methods generally suffer from poor generalization performance.   In this paper, we introduce BDrank, which could learn from brain dynamics preferences. Particularly, BDrank aims at preserving the ordering corresponding to the whole RTs instead of estimating the exact value of a single RT like previous attempts. Therefore, the main argument of this paper lies in Regression V.S. Ordinal Classification.
> SVR is chosen as the representative for the regression-based attempts for the following concerns: (1) SVR is considered to achieve the same performance with the shallow neural networks, while the parameters for SVR are much easier to tune to the best using cross-validation, as we did in the paper. (2) The number of trials for each participant is about 200, which is too small to train a deep neural network. (3) As is shown in Figure 3, even the SVR (using 5-fold cross validation) is still easier to overfit due to the non-smooth property of RTs, let alone other neural network based attempts.
> LOR is chosen as the representative for the ordinal classification based attempts due to its simplicity. LOR is optimized with L-BFGS with the default setting for all participants. The surprising result of LOR is a strong support to our claim that ordinal classification based attempts are more suitable for mental fatigue evaluation.  Of course, LOR can be easily replaced with more complex structures, e.g. neural network, to achieve more superior performance. Furthermore, LOR can also be extended to input image data, like fMRI.
> In a word, SVR and LOR are selected with careful consideration. The two baselines are considered to be a strong support for our claims: (1) Ordinal Classification based attempts are more suitable for mental fatigue evaluation, especially with non-smooth response variable (RTs); (2) the channel state indeed affects the performance of the learning model. Note that our BDrank, introducing a transition matrix on top of LOR, still enjoys the superiority for LOR.  Those extensions, which are not the focus of this paper, are left for later studies.
>
> Q2: The results are only somewhat interesting, no understanding of the underlying physiological processes is given.
> Page 2, Experiment Paradigm section mentioned that data recorded from the previous study from Huang et al 2015, showed that EEG related activity is correlated with behavioural information like reaction time (RT) and demonstrated that how the decline in vigilance occur during driving. The results presented in this paper further demonstrated that indeed behavioural information like RT is highly related to EEG brain dynamics.
> Further, Table 1 adopts the Wilcoxon-Mann-Whitney statistic to verify whether the learning model could maintain the pairwise comparison between two EEG signals (See Figure 1B). In terms of LOR and our BDrank, we just collect the prediction accuracy for each pairwise comparison and average on the whole dataset. In terms of SVR, we followed the standard procedure to train the learning model for each participant and then get the predicted RTs on the training and test dataset, respectively. Instead of calculating the RMSE loss, we calculate the classification error on the generated pairwise comparisons as what we do for LOR and BDrank.
> Table 2 and Figure 3 adopt the indegree sequence to verify whether the learning model could preserving the global ordering corresponding to RTs.  We first collected the indegree sequences (See Appendix) of the constructed directed graph using the predicted RTs and then measured the indegree discrepancy between the calculated indegree sequences and the ground truth using the root-mean-squared error (RMSE). See Figure 3, if the predicted indegree sequences could closely align with the ground truth, the learning model can perfectly preserve the whole ordering w.r.t. RTs.
> Figure 4 presents the estimated state of different channels for forty-four participants. Each column denotes the states of 33 channels for each participant. According to our analysis, the parameter \pi_n in the transition matrix indicates the channel reliability. Then we classify the 33 channels into positive (\pi_n>0.85), noisy (0.15<\pi_n<0.85) and negative (\pi_n<0.15) channels according to the estimated channel reliability \pi_n.

---

> > ### Author Response · Authors · 2018-11-13
> > **Our work is fundamental but not preliminary.**
> >
> > Q3: I consider the paper somewhat preliminary.
> > We do not agree with this statement. This work is fundamental but not preliminary. In this work, we proposed two claims: (1) learning from brain dynamic preferences; (2) channel state-aware within the learning model. And we resort to enough experiments to support our claims. In particular, in Table 1 and Table 2, LOR without fine-tuning achieves competitive performance with SVR, while the parameter of SVR is learned with 5-ford cross-validation. BDrank achieves better performance over LOR benefiting from the introduced transition matrix, which enables BDrank to only learn from reliable channels.
> > Our BDank is a basic model, which can be further extended to containing more complex structures when dealt with difficult situations, which are considered not the focus in this work.

---

### Official Review · AnonReviewer2 · 2018-11-05
**Work seems interesting but has limited theoretical advancement; appears to be out of scope for ICLR**

**Rating:** 4
**Confidence:** 3

**Review:**

The paper proposes an algorithm for mental fatigue monitoring, relating a subjects' EEG signals to their reaction time (RT) during a simulated driving task, as an ordinal regression problem. The authors argue that RTs could be heavily skewed and/or non-smooth, making traditional regression approaches unstable due to outlier values. They propose a brain dynamic ranking algorithm,  BDrank, using a generalized EM algorithm to estimate its parameters, and compare it to support vector regression and Logistic Ordinal Regression, where they show improved performance by accuracy and root mean squared error (RMSE) over a database of 44 subjects.

General comments, in no particular order:

1. There are some minor grammatical errors throughout. The paper could benefit from another read-through to correct these errors.

2. It is unclear to me how the model works at test time; as the model is essentially building a relational structure in the data, does the user have to provide multiple EEG trials at time of prediction?

3. There is notation early on in the paper that doesn't appear to be appropriately defined. For example, Equation (1) describes two sets of propositions, with M1 and M2 elements, respectively. How is M1 and M2, the total set of propositions, calculated? It appears to be all pair-wise comparisons of RTs but then it's unclear why there are two indexes associated with them. Also, what does it mean for the orderings to be significant? (i.e.: that the "type-1 preference propositions that the orderings between the RTs are significant"). The authors then switch to a new notation x^1 and x^2 without defining them. Notational problems also persist throughout the paper, making it hard to gauge what is being done at each step.

4. The authors describe using an FFT to transform the EEG data into the frequency domain. I'm assuming they are doing the FFT on the entire 10-s interval but the paper does not make this clear. Also, the authors state using EEG power between 0-30Hz for their analysis; do they further sub-divide this range (for example, to the standard theta/alpha/beta power ranges) or just use the power across the entire 0-30Hz band?

5. I am concerned with the relatively sparse set of comparison algorithms the authors use. The authors only compare to relatively simple approaches (support vector regression and logistic ordinal regression), yet they cite many previous works in this area but do not compare against them, instead just leaving a pretty generic statement of "The regression assumption of this method between EEG signals and RT is not correct"; they do not elaborate on this aspect.


Overall I think there is limited novelty in the approach; the idea to learn the structure of the data relationally instead of absolutely is pretty straight-forward, and is a standard practice for example in non-parametric statistical modeling. I am also not positive that ICLR is the best venue for this work; perhaps a better avenue for this would be in a more BCI/neural engineering-focused venue.

---

> ### Author Response · Authors · 2018-11-24
> **SVR and LOR are selected with careful consideration. The two baselines are considered to be a strong support for our claims**
>
> Thanks for your comments and suggestions regarding our paper. We summarize your comments into the following sub-problems and answer each one separately.
>
> Q1: It is unclear to me how the model works at test time; as the model is essentially building a relational structure in the data, does the user have to provide multiple EEG trials at time of prediction?
> A1: Yes, we need to refer to some baseline trials with known RTs when we do prediction. Through the pairwise comparisons with these trials using the learned model, we can get a coarse estimation of the RT. Furthermore, these baseline trials are easy to access as long as their RTs are diverse enough, e.g. some trails in the training dataset with diverse RTs.
>
> Q2:  Equation (1) describes two sets of propositions, with M1 and M2 elements, respectively. How is M1 and M2, the total set of propositions, calculated? It appears to be all pair-wise comparisons of RTs but then it's unclear why there are two indexes associated with them.
> A2: D1 denotes the type-1 preference propositions that the orderings between the RTs are significant. M1 is the total number of D1; while D2 denotes the type-2 preference propositions that the RTs in each comparison are comparable.  M2 is the total number of D2. We use different indexes for D1 and D2 for the convenience of explanation in the optimization part.
>
> Q3: What does it mean for the orderings to be significant? (i.e.: that the "type-1 preference propositions that the orderings between the RTs are significant").
> A3: The word "significant" denotes the difference between RTs is significant, namely RT1 is significantly greater/smaller than RT2, regardless of random unknown noisy introduced by the instrument error. We empirically realize this as RT2 < min (RT2+1, 1.5*RT2) < RT1 in the experiment part.
>
> Q4: The authors then switch to a new notation x^1 and x^2 without defining them.
> The pairwise brain dynamics preference (x^1_{n,m}, x^2_{n,m}) is first introduced after Equation 1,  denoting the features recorded within the n-th channel for each preference proposition in D1 or D2. In Section 2b, we omitted the subscript (n,m) for the sake of simplicity.
>
> Q5: The authors describe using an FFT to transform the EEG data into the frequency domain. I'm assuming they are doing the FFT on the entire 10-s interval but the paper does not make this clear. Also, the authors’ state using EEG power between 0-30Hz for their analysis; do they further sub-divide this range (for example, to the standard theta/alpha/beta power ranges) or just use the power across the entire 0-30Hz band?
> A5: Yes, Your understanding is correct. We do the FFT on the entire 10-s interval. And we use the power across the entire 0-30Hz band as the feature vector.
> If needed, we can further introduced the group loss on the regression weight w, with each group corresponding to the standard theta/alpha/beta power ranges, respectively. Let our model to select the most relevant power ranges in a data-driven approach.
>
> Q6: I am concerned with the relatively sparse set of comparison algorithms the authors use. The authors only compare to relatively simple approaches (support vector regression and logistic ordinal regression),
> A6: SVR and LOR are selected with careful consideration. The two baselines are considered to be a strong support for our claims: (1) Ordinal Classification based attempts are more suitable for mental fatigue evaluation, especially with non-smooth response variable (RTs); (2) the channel state indeed affects the performance of the learning model.
>
> SVR is chosen as the representative for the regression based attempts for the following concerns: (1) SVR is considered to achieve the same performance with the shallow neural networks, while the parameters for SVR are much more easier to tune to the best using cross validation, as we did in the paper. (2) The number of trials for each participant is about 200, which is too small to train a deep neural network. (3) As it shown in Figure 3, even the SVR (using 5-fold cross validation) is still easier to overfit due to the non-smooth property of RTs, let alone other neural network based attempts.
>
> LOR is chosen as the representative for the ordinal classification based attempts due to its simplicity. LOR is optimized with L-BFGS with the default setting for all participants. The surprised result of LOR is a strong support to our claim that ordinal classification based attempts are more suitable for mental fatigue evaluation.  Of course, LOR can be easily replaced to more complex structures, e.g. deep neural network, to achieve more superior performance. Furthermore, LOR can also be extended to input image data, like fRMI. Note that our BDrank, introducing a transition matrix on top of LOR, still enjoys the superiority of LOR.  Those extensions, which are not the focus of this paper, are left for later studies.

---

> > ### Author Response · Authors · 2018-11-24
> > **The novelty of our work lies in the view of the paper which we model the mental fatigue and also the potential impact of the work on the later research.**
> >
> > Q7: They cite many previous works in this area but do not compare against them, instead just leaving a pretty generic statement of "The regression assumption of this method between EEG signals and RT is not correct"; they do not elaborate on this aspect.
> > A7: In the following, we elaborate our statement about the deficiency of the regression assumption from two aspects: theoretical analysis and empirical experiment.
> >
> > First, the extreme RTs are widely existed during data collection, which means RTs are unevenly distributed over the positive real numbers R+. This is the reason why we call the response variable (RTs) are non-smooth. According to function approximation theory, it usually induces us to use over-complex structure so as to approximate the exact value of the extreme response time. The over-complex structure would further lead to the overfitting of the learning model.
> >
> > Second, our statement was empirically verified by the experiment. Table 1 and Table 2 collect the accuracy of DBrank and other two baselines on the training and testing dataset using two different metrics, it consistently supports our claim that regression based method (SVR) is easy to overfitting to non-smooth response variables (RTs), while ordinal classification based attempts (LOR and BDrank) are robust. Figure 3 gives a more intuitive presentation:  SVR usually fluctuates heavily for low indegree trials (denoting small RTs) and high indegree trials (denoting large RTs). It means that SVR over-estimates the RTs with small values and under-estimates the RTs with large values.
> >
> > Q8: The idea to learn the structure of the data relationally instead of absolutely is pretty straight-forward, and is a standard practice for example in non-parametric statistical modelling.
> > A8: This statement is quite subjective. The novelty of our work lies in the view of the paper which we solve the problem and also the potential impact of the work on the later research. Learning from structure could be standard practice but at the same time, nobody in our knowledge applied these techniques for physiological data (particularly for EEG data). The main novelty of method to apply on the EEG data, which also assume to work by comparison/preferences. Such that, brain always compare and prefer on option over other and our proposed method also uses a similar methodology to learn from data, which enhance the performance of method significantly. Overall, we aim to build a learning model which is robust to non-smooth response variables (RTs) and also robust to noisy channels. BDrank is the first work which could address these two challenges in the same model, by learning brain dynamics preferences only from informative EEG channels.
> >
> > Q9: I am also not positive that ICLR is the best venue for this work; perhaps a better avenue for this would be in a more BCI/neural engineering-focused venue.
> > A9: ICLR may be not the best but still one of the potential venue for our work. One of first reason is that, ICLR welcomes the topics about the applications in neuroscience, as listed in the Call for Papers. Second, ICLR is interested in the choices about data representation. We propose to learn from brain dynamics preferences, which is different from previous data representation paradigm. Furthermore, a transition matrix is introduced to characterize the reliability of each channel, which can be also viewed as the data representation of the channel state. As for the (deep) neural network extension of this work, it may be a potential extension when the training samples are sufficient.

---

### Official Review · AnonReviewer3 · 2018-11-06
**Review of mental fatigue monitoring using brain dynamics preferences**

**Rating:** 7
**Confidence:** 3

**Review:**

The mental fatigue is an important factor in road accidents. Finding a direct mapping between EEG features and reaction time is difficult and error-prone, combining the noise measurement of EEG and individual variation of RT.  The authors introduce a measure called BDrank based on partial ordering instead of regression. Formulating the measure as a MAP problem, the authors propose a generalized EM algorithm for prediction. An online extension, relying on iterative L-BFGS optimization over mini-batches.

Figure 3 shows the indegree sequence for 4 selected subjects. What is the criterion to select these subjects? These cases seem interesting, but is it representative for the best/worst case? It could provide some information to show some of the few cases where SVR is more accurate than BDrank.
Regarding the identification of noisy channels, the 33rd channel is indicated as a non-EEG one. What is it?

Some minor questions and suggestions:
- It could be interesting to mention the performance of this measure using only a limited set of EEG channels to evaluate its robustness.
- The introduction indicates that de Naurois et al ., 2017 rely on EEG to estimate the RT, but it is not the case.
- The formulation of the assumption (2) on page 3 is unclear, as sensors are not supposed to make any emission and there is a high correlation between channels.
- The model do not consider transition between type-1 and -2 preference, could it be a problem with confidence interval

---

> ### Author Response · Authors · 2018-11-26
> **Thank you so much for being an inspiration. It helps us to develop a better understanding of our problem and our BDrank model.**
>
> Q1: Figure 3 shows the indegree sequence for 4 selected subjects. What is the criterion to select these subjects? These cases seem interesting, but is it representative for the best/worst case?
> A1: As we already mentioned in the paper, we selected participants P19, P38, P42, P43 with the most representative (best) performance (See Figure 3). Although we only present the results of the participants with the best performance, we insist the statement also applies to other participants for the two reasons: (1) Figure 3 is just the visualization of Table 2. Therefore, similar observations can be found on the results of 39 participants. (2) The performance of SVR is calculated with the optima parameters for each participants, while fixed parameters are adopted for all participants in the experiment of LOR and BDrank. It means the performance for BDrank can be further improved by fine tuning the parameter for each participant.
>
> Q2: It could provide some information to show some of the few cases where SVR is more accurate than BDrank.
> A2: Thank you very much for the suggestion.
> We have investigated the principle behind the statistics of the participants which could achieve superior performance using SVR instead of BDrank. The two metrics on P14, P21, P23, P32, P33 using SVR are superior to BDrank. The reaction time of P14 and P21 indeed have many extreme values, but these extreme values are even distributed, which can be regarded as kind of smoothness. The reaction time of P32 and P33 even do not have too extreme values, and there are among 300 trials completed by these participants. P23 have only one extreme value but it has sufficient (334) trials.
> In the following, we summarize the situations where SVR can achieve comparable or better performance: (1) no or a small number of extreme values; (3) even distributed extreme values; (3) sufficient trials a.k.a. training samples.
> It is interesting to note that the above conclusion also consistent with our claim that the regression-based model is not suitable for the tasks with non-smooth response variable.
>
> Q3: Regarding the identification of noisy channels, the 33rd channel is indicated as a non-EEG one. What is it?
> A3: This is the channel receiving the information about vehicle position with the same sampling rate as EEG channels and contains the data about only one axis in the direction of deviation. This is the reason we called it non-EEG channel.
>
> Q4: It could be interesting to mention the performance of this measure using only a limited set of EEG channels to evaluate its robustness.
> A4: Thanks for your suggestion. Actually the current experiment results can provide us some preliminary results for the experiment the reviewer suggested. There are two kinds of parameters in the proposed BDrank model, namely the regression weight w and channel reliability \pi_n, n = 1, 2, ..., 33. In the following, we give our detailed analysis to support our claims that the number of the EEG channels do not affect the performance of our BDrank model.
> (1) The estimation of two parameters is independent from the number of EEG channels. In terms of w, it is a low dimension (493*1) vector. Therefore, it could be estimated very well using the EEG signals from limited or only one channels. Note that BDrank degenerates to LOR when learning from single channel.
> In terms of \pi_n, it is a scalar introduced for each channel. It could be trained well as usual as long as the corresponding channel is used.
> (2) The two parameters could still be estimated well when a considerable proportion of channels are noisy channels. See Fig.4, we list the reliability of different channels for forty four participants. Note that P32, P35, P42 could still achieve very high performance (ACC > 77%, in Table 1) when about one third of channels are recognized as noisy channels (the noisy channels are automatically eliminated from the training process by our BDrank.).
> In summary, our BDrank is robust to the number of channels as long as majority of channels are reliable (positive or negative).
>
> Q5: The introduction indicates that de Naurois et al ., 2017 rely on EEG to estimate the RT, but it is not the case.
> A5: That’s correct, this work is derived from behaviour information (expert rating) of drowsiness level which is highly correlated related with RT but less noisy information. Therefore, de Naurois et al ., 2017 was listed under methods related to RT. We agree with the reviewer to separate that reference with more details.

---

> > ### Author Response · Authors · 2018-11-26
> > **Thank you very much for the suggestion.**
> >
> > Q6: The formulation of the assumption (2) on page 3 is unclear, as sensors are not supposed to make any emission and there is a high correlation between channels.
> > A6: All EEG channels were spatially located over the scalp in the 10-20 International system. Although all of them receiving mix signals from all parts of brain therefore highly correlated. However, our assumption is based on each EEG channel is receiving and recording signal independently as system point of view with a specific sampling rate. Therefore, we emphasize in assumption 2 on page 4 that 33 EEG channels are independent and identically distributed w.r.t each trail.
> >
> > Q7: The model do not consider transition between type-1 and -2 preference, could it be a problem with confidence interval.
> > A7: This is a good question. Cause effects are two sides of the generalized transition matrix with one deriving a more complete but complex model and the other side is more model parameters and less interpretability.
> > Particularly, considering generalizing the transition matrix, it needs be reminded that the sum of each raw of the transition matrix be one. Therefore, at least three parameters need to be introduced. Fortunately, the resultant transition matrix can still be solved with EM algorithm. Note that the generalized transition matrix is similar to the state transition matrix in HMM, and most solution for HMM can be extended to our model. However, the concise but useful interpretation of the latent variable \pi_n no longer exists, since the channel state is now characterized by more than two parameters.
> > For our current application, this extension would bring marginal performance improvement but definitely not improve human interpretability.

---

### Meta-Review · Area_Chair1 · 2018-12-16

**Confidence:** 4
**Recommendation:** Reject

**Metareview:**

The manuscript describes a novel technique predicting metal fatigue based on EEG measurements. The work is motivated by an application to driving safety. Reviewers and the AC agreed that the main motivation for the proposed work, and perhaps the results, are likely to be of interest to the applied BCI community.

The reviewers and ACs noted weakness in the original submission related to the clarity of the presentation and breadth of empirical evaluation. In particular, only a few baselines were considered. As a result, for the non-expert, it is also unclear if the proposed methods are compared against the state of the art. There was also a particular concern that this work may not be a good fit for an ICLR audience.